# GIS-based land suitability evaluation and multi-criteria decision analysis for sustainable enset (Ensete ventricosum (Welw.) Cheesman) cultivation in Hadiya Zone, Central Ethiopia

**Alemu Ersino Ersado**[1]*, **Venkata Krishna Talluri**[2]

**1** Geography Department at the College of Science and Technology, Andhra University, Visakhapatnam, Andhra Pradesh, India, **2** Geography Department, College of Science and Technology, Andhra University, Visakhapatnam, Andhra Pradesh, India

\* alemuersino@gmail.com

## Abstract

Land suitability analysis is a key approach for evaluating the potential of land resources for specific uses and for supporting sustainable agricultural planning. In Ethiopia, where agriculture forms the backbone of rural livelihoods, identifying suitable land for staple crops is essential to ensure food security and long-term productivity. This study evaluated the actual land suitability for enset (Ensete ventricosum) cultivation in the Hadiya Zone, Central Ethiopia, by systematically comparing the spatial distribution of key environmental factors with established enset crop requirement standards. For each parameter, spatial data were overlaid with enset-specific ecological thresholds derived from relevant literature and expert consultation. Based on the FAO land evaluation framework, all factors were classified into five suitability classes: Very Highly Suitable (S1), Highly Suitable (S2), Moderately Suitable (S3), Marginally Suitable (N1), and Permanently Not Suitable (N2), enabling the identification of spatial variability in enset suitability and supporting subsequent multi-criteria evaluation and weighted overlay analysis. The analysis evaluated criteria such as soil properties (type, depth, organic carbon content, pH, and texture), topographic situation (slope and elevation), climate variables (rainfall and temperature), and LULC. The integrated analysis revealed that enset cultivation is highly favorable across most of the study area, with 57.72% classified as highly suitable (S1), 36.89% as moderately suitable (S2), 0.16% as marginally suitable (S3), and 5.23% as currently not suitable (N1), while no areas were identified as permanently unsuitable (N2). Overall, the results highlight the strong natural potential of the Hadiya Zone for enset cultivation, although localized constraints related to soil fertility, water availability, and slope conditions may require targeted management interventions.

**Data availability statement:** All data generated or analyzed during this study are included in this publishing article. Additional data can be provided by the corresponding author upon reasonable request.

**Funding:** The author(s) received no specific funding for this work.

**Competing interests:** The authors declare that they have no known competing financial interests or personal relationships that could have appeared to influence the work reported in this paper.

## Introduction

Agriculture remains one of the most fundamental human activities, sustaining livelihoods and ensuring food security worldwide. Although global agriculture continues to demonstrate the capacity to overtake growing food demand [1], food insecurity remains a pressing challenge in Sub-Saharan Africa. In the past three decades, agricultural production in Sub-Saharan Africa has deteriorated, failing to increase per capita daily calorie supply beyond 2,100, while also losing export competitiveness [2,3]. Rapid population growth, limited arable land, and land degradation continue to intensify pressures on the agricultural system [4]. Addressing these challenges requires aligning crops with the environments best suited for their growth. Land suitability analysis, therefore, plays a crucial role in enhancing food production, optimizing resource use, and ensuring long-term sustainability [4]. Given the finite and non-renewable nature of land resources, careful allocation is necessary to balance competing demands.

Land use competition among agriculture, urban development, and preservation demands plans that consider economic, environmental, and social dimensions to promote sustainable development [5,6,7]. The challenge, therefore, lies in creating a land use policy that not only acknowledges these competing demands but also attempts to find a balance that promotes equitable and sustainable development across all sectors. This involves a complex interplay of economic, environmental, and social considerations to optimize land use in a way that supports current and future generations [8].

According to some scholars, analysis of land suitability is very needed to contribute to the world's food production in general and in particular in Ethiopia, to improve food security [9]. The process of land suitability analysis is the categorization and grouping of specific areas of land in terms of their suitability for defined usage [10,11]. Land needs careful and appropriate use to achieve optimum productivity and to ensure environmental sustainability for future generations. This requires effective and operative management of land information on which such decisions should be based because land is one of the non-renewable natural resources, [7]. The decision on appropriate use includes the past and present human activities and the status of the physical and chemical properties of the land. Land evaluation is concerned with the assessment and valuation of land when used for specified purposes [12,13].

The challenge of identifying suitable land for the development of specific agricultural products is a longstanding and predominantly empirical concern [9]. Despite extensive research on agricultural productivity and food security globally, the detrimental effects of poor land resource management and the failure to utilize land according to its optimal suitability remain significant challenges, especially in developing countries.

Land suitability evaluation categorizes land according to its appropriateness for particular applications. Suitability is contingent upon crop requirements and land features, assessing the alignment of land qualities with specific uses. The primary objective is to forecast the capacity of land units for sustainable applications without resource deterioration [14]. Physical land evaluation is crucial for land-use planning,

as it directs optimal resource utilization. In multi-criteria decision-making, many land criteria are assigned weights to identify the optimal land use [15]. Consequently, land might be designated appropriateness levels to govern optimal utilization. GIS-based suitability analysis, integrating decision-makers' preferences, provides enduring solutions for locating productive land [1]. In Central Ethiopia, inappropriate utilization of land has resulted in significant issues, although their magnitude has not been adequately researched [16].

Conducting a physical land suitability analysis in the Hadiya zone is essential, as agriculture, predominantly rain-fed, serves as the principal livelihood for the majority of populations [17]. Rapid population expansion, constrained livelihood opportunities, and climate variability are causing a decline in agricultural production per hectare, resulting in prolonged food insecurity. Consequently, assessing land suitability by multi-criteria parametric approaches is essential for optimizing land potential and guaranteeing sustainable agricultural productivity, [18,19].

This study addresses the essential gap by highlighting the necessity for a systematic land suitability analysis for enset agriculture. Despite being a crucial staple crop, enset production is frequently omitted from formal land-use planning and predominantly depends on local knowledge. A scientifically grounded land suitability evaluation is critically needed to improve productivity and promote sustainable resource management despite increasing land degradation, population pressure, and climatic variability [20,21]. Cropland suitability analysis correlates land capability with the specific requirements of crops. This study delineates appropriate areas for enset cultivation and analyses the principal environmental elements affecting its appropriateness. Although several crop suitability studies have been undertaken in Ethiopia, enset has attracted insufficient focus, especially in the Hadiya Zone in Central Ethiopia, where it is fundamental to local lives.

This research was carried out in the Hadiya Zone of Central Ethiopia, an area known for its various agro-ecological conditions, varied topography, and intensive land-use patterns. The study combines Geographic Information Systems (GIS) and Multi-Criteria Decision Analysis (MCDA) to assess and map land suitability for enset production. The analysis takes into account essential biophysical and environmental aspects such as physical and chemical soil properties, elevation, slope, climate, and land use/cover [6]. The study's comprehensive methodology aims to create a clear spatial picture of land suitability for enset farming, enabling informed decision-making for sustainable agricultural planning and resource management [4]. To fill a critical research gap, the study uses a GIS-based Analytical Hierarchy Process (AHP) to evaluate whether the site is suitable for enset farming in the Hadiya Zone. Because no previous GIS-AHP-based suitability evaluation for enset has been conducted in the research area, the findings are expected to be useful for agricultural planners, development practitioners, and policymakers.

### Specific Objectives

1. To identify the key factors used in assessing land suitability for enset cultivation.

2. To quantify the relative influence of different factors on enset cultivation suitability.

3. To classify and map land units according to their suitability levels for enset cultivation.

4. To evaluate the overall land suitability of the study area for enset cultivation using GIS and multi-criteria decision analysis.

## Materials and methods

### Description of the study area

Fig 1 illustrates the location and agro-climatic characteristics of the Hadiya Zone in Central Ethiopia. According to Ethiopia's simplified agro-climatic classification, the zone covers approximately 3,635.5 km$^2$ and comprises three distinct agro-climatic zones: the temperate and cold highlands (Dega or *Hansawa*, 2500–3200 m a.s.l.), the warm midlands (Woina Dega or *Hansaw Kaala*, 1500–2500 m a.s.l.), and the hot lowlands (Kola or *Qaala*, 500–1500 m a.s.l.), accounting

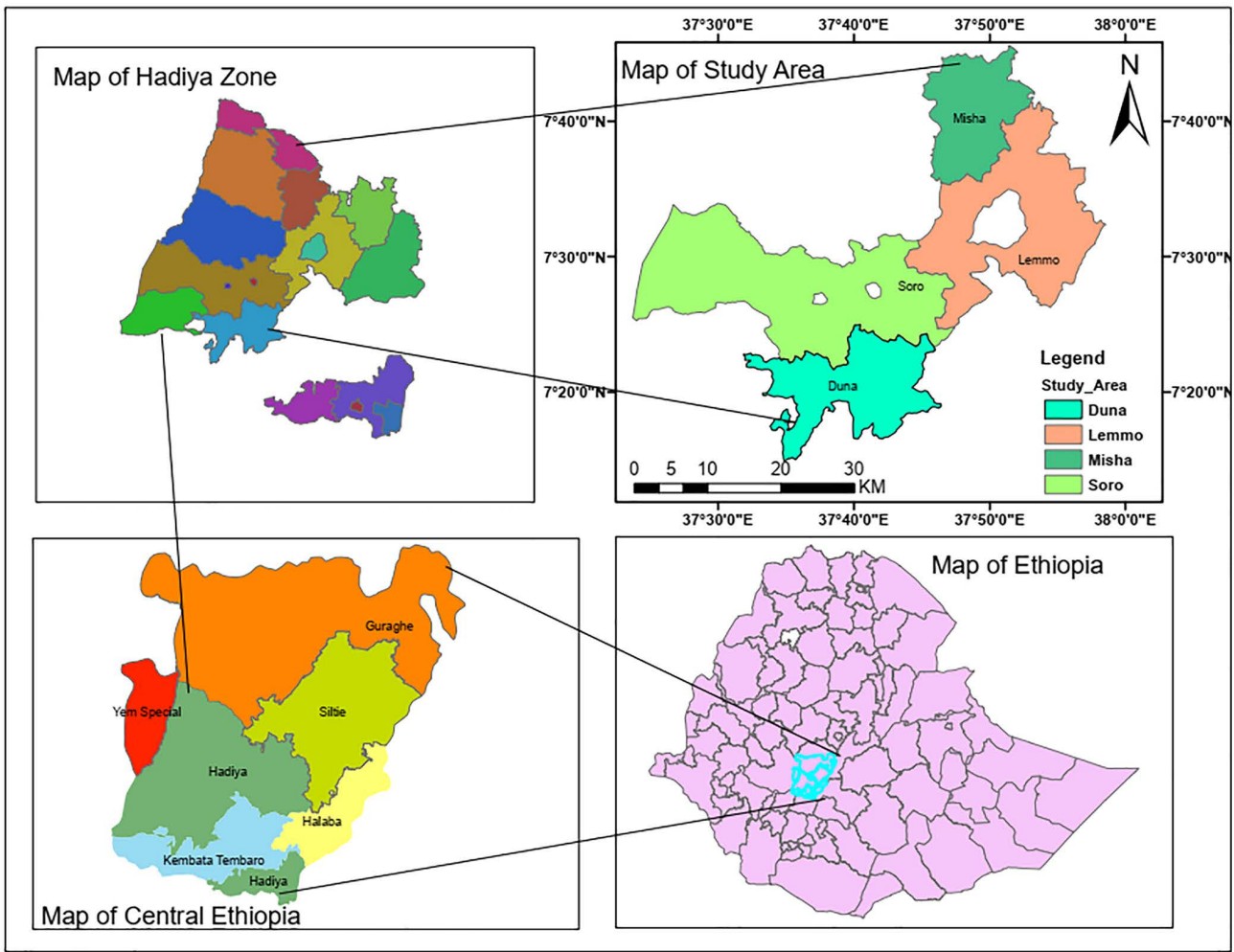

**Fig 1. Map of the study area.**

for 18%, 69.1%, and 12.9% of the total area, respectively [22]. Within the sampled districts, Misha and Duna are classified as Dega, while Lemo and Soro fall within the Woina Dega agro-climatic zone.

The study area is located in the Hadiya Zone, situated on the western margin of the Central Rift Valley, in the northwestern part of the former Southern Nations, Nationalities, and Peoples' Regional State (SNNPR). The zone is bounded by the Oromia Region and Yem Zone to the northwest, Halaba and Silte Zones to the east and northeast, and Kembata Zone and Tembaro Special Woreda to the south. To the north and northwest, it borders the Gurage Zone and Kebena Special Woreda, while the detached woredas of Misrak Badawacho, Mirab Badawacho, and Siraro share boundaries with the Wolaita Zone and Oromia Region. Geographically, the zone extends from 7°05′ to 8°32′ N latitude and 37°29′ to 38°13′ E longitude, lying entirely north of the equator and east of the prime meridian. Owing to its position in the tropical highlands, the Hadiya Zone experiences a climate that resembles mid-latitude conditions, where altitude and topography play a dominant role in shaping local climatic and environmental patterns, thereby exerting a significant influence on agricultural systems and livelihoods [13].

Administrative boundary data for Ethiopia were obtained from the Humanitarian Data Exchange (HDX) dataset "Ethiopia Administrative Boundaries (COD-ABETH)," provided under the Creative Commons Attribution 4.0 International (CC BY

4.0) license. Climate data used for spatial mapping were obtained from the World Bank Climate Knowledge Portal, which provides open-access climate datasets derived from publicly available global and national sources. These datasets were processed, analyzed, and mapped by the authors.

## Methodology and materials

This study investigates how land suitability for sustainable enset cultivation in the Hadiya Zone, Central Ethiopia, is assessed using an integrated Geographic Information System (GIS) and Multi-Criteria Decision Analysis (MCDA) approach. Key biophysical factors, including soil properties, topography, climate, and land use/land cover, were standardized and weighted using the Analytical Hierarchy Process (AHP). Weighted overlay analysis was applied to generate land suitability classes and assess their implications for sustainable agricultural production and livelihood improvement.

As indicated in Table 1, to achieve the objectives of the study, both primary and secondary data sources were utilized. Significant datasets included high-resolution soil maps, Landsat 8 satellite imagery, Digital Elevation Model (DEM) data for elevation and slope analysis, and long-term averages of temperature and precipitation. These inputs were selected based on their relevance to determining the ecological suitability of land for enset cultivation and were processed and analyzed using GIS and Multi-Criteria Decision-Making (MCDM) techniques.

## Methods of data processing and analysis

Agricultural land appropriateness includes a variety of technological, environmental, social, and economic factors. As shown in Fig 2, the GIS-based MCDA approach analyses and combines spatial data with decision-makers' preferences to generate decision-making information. The decision-maker can receive varying aid in choosing a suitable location by utilizing the special approaches that GIS and MCDA provide to organize and integrate a wide range of data that are examined in a variety of methods. The Analytical Hierarchy Process (AHP) was used to give relative weights to each criterion, followed by a weighted overlay approach to construct the final suitability map [8].

**Table 1. Source and types of data.**

| Data Type | Data Source | Resolution/Format | Software's used |
|---|---|---|---|
| Climate Data | National Meteorological Agency (NMA), Ethiopia World Bank Climate Knowledge Portal https://climateknowledgeportal.worldbank.org/country/ethiopia | Average monthly/annual temperature and rainfall data (station-specific). ERA5: 0.25° (~31 km), Raster (1 km$^2$) | ArcGIS 10.8 |
| Digital Elevation Model (DEM) | Open Topography (https://opentopography.org/) | SRTM: 30m raster | ArcGIS 10.8, ERDAS Image 2015 |
| Soil Data | Fao Soil Dabase; Africa Soil profiles Database (Afsp); Field soil survey | Raster:250m resolution; Vector/Raster; laboratory-analyzed soil physico-chemical data | ArcGIS 10.8 |
| Land Use and Land Cover (LULC) Sentinel 2 | Copernicus Global Land Service Land Cover (https://land.copernicus.eu/en/products/global-dynamic-land-cover) | 10m spatial resolution; Raster (2022–2024 images) | ArcGIS 10.8 |
| Satellite Imagery (Raster) Landsat 8 OLI (2024) | USGS (www. earthexplorer.gov) | 30 m × 30 m; Multispectral raster | ArcGIS 10.8 |
| Ground Truth/ Field Data | GPS survey; farmer interviews; expert judgment | Point data; attribute tables | ArcGIS 10.8 |
| Administrative Boundaries | Humanitarian Data Exchange (HDX) https://data.humdata.org/dataset/cod-ab-eth | Vector (Shape file) | ArcGIS 10.8 |

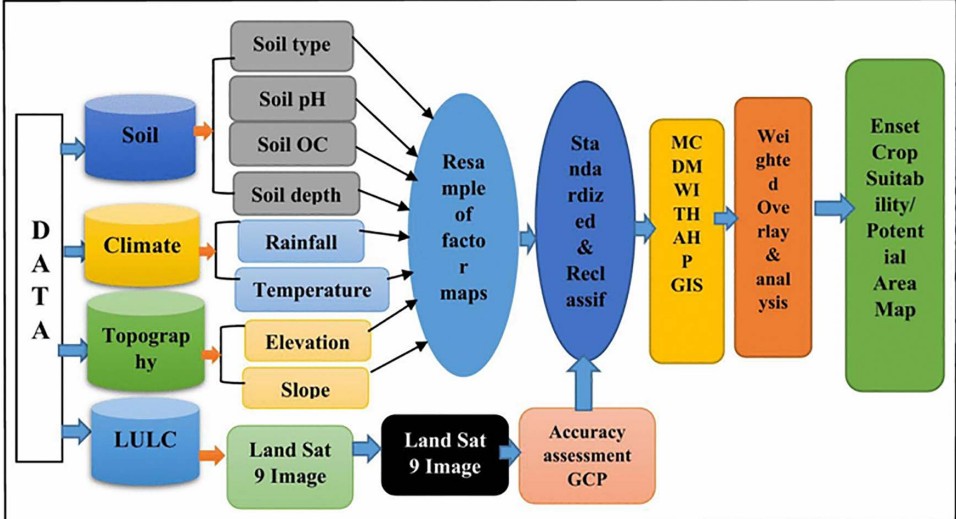

**Fig 2. Flow diagram of the method used for suitability analysis.**

## Ethics approval and consent to participate

Ethical approval was not required for this study as it did not involve human participants or animals. Or Clinical trial number: not applicable.

## Results and discussions

### Spatial land suitability analysis for enset cultivation

**Computation of the pairwise comparison matrix and consistency.** A pair-wise comparison matrix is created to assign weights by experts. Weights are evaluated to find alternatives and estimate associated absolute numbers from 1 to 9 in the fundamental scales of AHP presented in Table 2. For determining the relative importance of criteria, the pair-wise comparison matrix using Saaty's nine-point weighting scale was applied [13].

**Estimation of the consistency ratio.** To construct the pairwise comparison matrix, the maximum eigenvalue (λmax) and the Consistency Index (CI) were calculated. The value of λmax was obtained as the average of the consistency vector. To minimize subjectivity and prevent bias in criteria weighing, the Consistency Ratio (CR) was computed to assess the reliability of the judgments.

$$C.I. \ = \frac{\lambda_{max} - n}{n - 1}$$

(1)

$$C.R. \ = \frac{C.I}{R.I}$$

(2)

Where: n is the number of items being compared in the matrix

$\lambda_{max}$ = The largest eigenvalue or WSV/CW

- WSV = weight sum value

- CV = criteria Weight

**Table 2. Nine-point Weighting Scale for Pair-wise Comparison.**

| Intensity of Relative Importance | Scale |
|---|---|
| Equally Important | 1 |
| Equal to moderate importance | 2 |
| Moderate importance | 3 |
| Moderately to strongly | 4 |
| Strong importance | 5 |
| Strong Plus/ | 6 |
| Very Strong importance | 7 |
| Very Strongly Plus | 8 |
| Extreme Importance | 9 |

R.I. = random consistency

C.I. Consistency Index

For the present analysis:

$\lambda_{max} - n = 10.458 - 10 = 0.458$

n-1 = 9

CI =) 0.051

RI = 1.49

**CR = <u>0.0342 or 3.4%</u>**

The Consistency Ratio (CR) was calculated using Expert Choice software. The resulting CR value (0.0342, or 3.4%) is well below the commonly accepted threshold of 0.10 (10%), indicating a high level of consistency in the pairwise comparison matrix. Consequently, no further revision of the judgment matrix was required.

### Factors of enset cultivation suitability analysis based on

**Land use/land cover classification.** As indicated Table 3, the Land Use/Land Cover (LULC) classification of the study area was carried out to identify the spatial distribution of various land cover types and assess their suitability for enset cultivation. The classification process involved organizing image pixels into predefined classes based on their spectral characteristics, supported by representative training sites that defined the unique spectral and statistical signatures of each class. These signatures were applied across the imagery to produce a thematic LULC map. According to some research, delineating the existing land-use boundary was the first step in the land evaluation process, [23] and [19].

**Table 3. LU/LC classes and its suitability of the study are.**

| LULC classes | Level of suitability | Value | Area (Km²) | Area coverage (%) |
|---|---|---|---|---|
| Crop Land | Highly Suitable | S1 | 1879.7484 | 51.7 |
| Grass Land/ Range Land/Wet Land/Bare Land | Moderately Suitable | S2 | 569.8516 | 15.67 |
| Bush Land/ Wood Land | Marginally Suitable | S3 | 386.8 | 10.6 |
| Forest | Currently not Suitable | N1 | 736.9 | 20.3 |
| Built-up/Water | Permanently not Suitable | N2 | 62.3 | 1.73 |

Figs 3 and 4 indicate that cropland is the dominant land cover type, covering 1,879.75 km² (52%) of the study area, and is classified as highly suitable for enset cultivation due to favorable soil properties, topography, and moisture availability. Grassland/rangeland occupies 569.85 km² (15.67%) and is moderately suitable, primarily because of relatively lower soil fertility and water retention capacity. In contrast, forest areas (736.9 km², 20.3%) and built-up areas (62.3 km², 1.7%) are considered unsuitable, as their land cover characteristics restrict agricultural use. Overall, the findings highlight cropland as the most promising land cover for the expansion of enset cultivation, while also identifying areas where cultivation is constrained by land cover limitations [24].

**Accuracy assessment interpretation.** The land-use/land-cover categorization attained an overall accuracy of 89.7% and a Kappa coefficient of 0.7905, indicating substantial agreement beyond mere chance. Class-specific accuracies were consistently elevated: Producer's Accuracy varied from 94.7% for farmland to 83.3% for built-up/water, whereas User's Accuracy ranged from 95.7% for settlement to 80% for bushes. This implies a few omissions and commission mistakes, predominantly below 15%. Overall, the classification is reliable for mapping major land-use/land-cover types and provides a solid basis for further analyses like land suitability evaluation [[21] see Table 4.

$$\text{1. Overall accuracy (OA)} = \frac{Sum\ of\ diagonal}{Total\ samples} = 0.8966 \text{ or } 89.7\%$$

$$\text{2. Kappa coefficient (k)} = \frac{P_0 - P_e}{1 - P_e} = 0.7905 \text{ or } 79.1\% \text{ acceptable.}$$

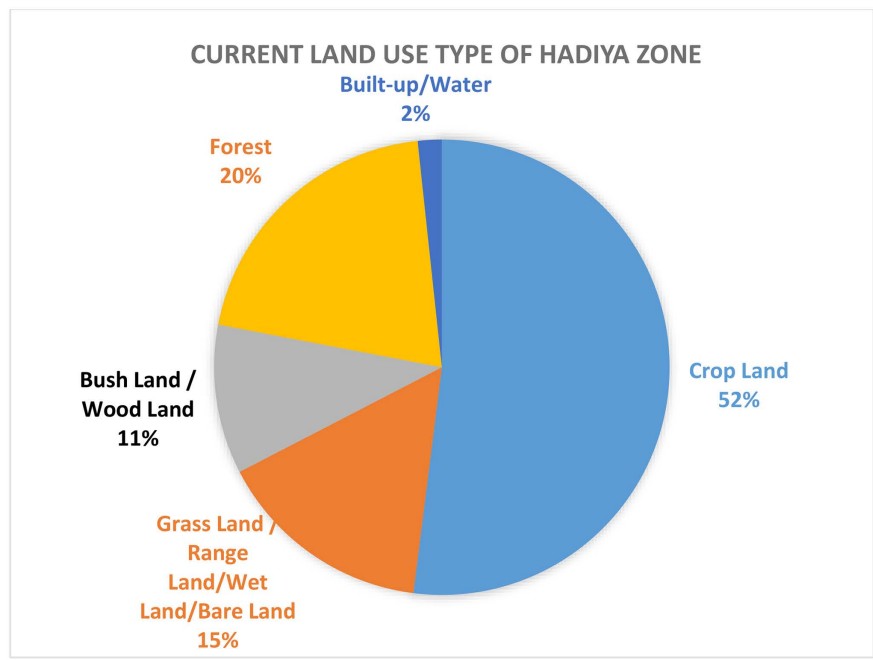

**Fig 3. Current reclassified land use type in the study area.**

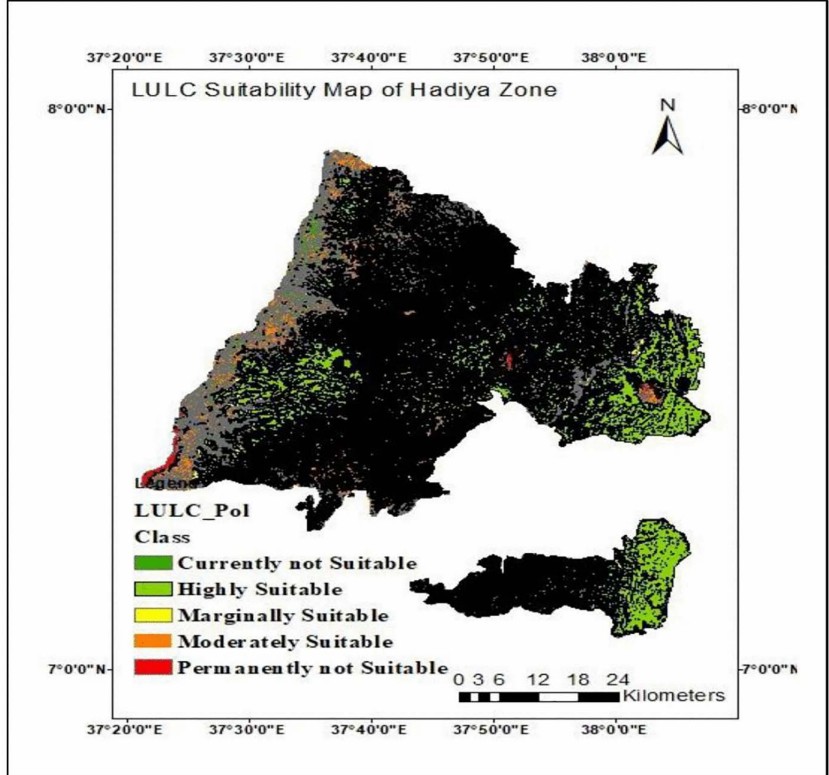

**Fig 4. LU/LC classes of the study area.**

**Table 4. Accuracy assessment of Landsat 8 2022 Classification.**

| Classified Data | Reference Data | | | | | | | Row Total | Number Correct | Producers Accuracy (%) | Users Accuracy (%) |
|---|---|---|---|---|---|---|---|---|---|---|---|
| | Farmland | Forests | Bushes | Grasslands | settlement | Bared land | Water | | | | |
| Farmland | 36 | 0 | 1 | 1 | 0 | 0 | 0 | 38 | 36 | 94.7 | 92.3 |
| Forests | 1 | 24 | 1 | 1 | 0 | 0 | 0 | 27 | 24 | 88.9 | 92.3 |
| Bushes | 0 | 1 | 12 | 1 | 0 | 0 | 0 | 14 | 12 | 85.7 | 80 |
| Grasslands | 1 | 1 | 0 | 18 | 0 | 0 | 0 | 20 | 18 | 90 | 85.7 |
| Settlement | 1 | 0 | 0 | 0 | 22 | 0 | 1 | 24 | 22 | 91.7 | 95.7 |
| Bared land | 0 | 0 | 1 | 0 | 0 | 8 | 0 | 9 | 8 | 88.9 | 88.9 |
| Water/Built up | 0 | 0 | 0 | 0 | 1 | 1 | 10 | 12 | 10 | 83.3 | 90.9 |
| CT | 39 | 26 | 15 | 21 | 23 | 9 | 11 | 145 | 130 | – | – |

## Enset suitability analysis based on topography

Elevation and slope were two topographic factors taken into account in this study, which were analyzed in the ArcGIS environment as follows:

**Elevation.** Based on the simplified agro-climatic classification of Ethiopia, the zone has three agro-ecological zones, namely Kola or semi-desert (lowland < 1500 m) covering about 12.9% of the land area, Woina Dega or cool semi-arid

(mid-altitude 1500–2500 m) about 69.1% and Dega or cool and humid (highland > 2500 m) about 18% [25]. Most of the area lies within the mid-altitude zone [26]. The altitude of the study area ranges from 2961 to 727 masl. Topography significantly affects plant growth as it determines yield variability. Natural features, elevation gradients, and elevation classification greatly influence enset yield and quality [27].

According to Table 5 elevation is a key determinant of land suitability for enset cultivation in the Hadiya Zone. Areas situated above 2100masl which account for 29.7% of the zone, are classified as highly suitable. Elevations between 2100 and 1900 m (44.6%) are moderately suitable, while areas ranging from 1900 to 1700 m (14.4%) are marginally suitable. In contrast, land between 1700 and 1500 m (5.3%) is currently unsuitable, and areas below 1500 m (6.0%) are considered permanently unsuitable. Overall, nearly 89% of the zone lies above 1700 m a.s.l., indicating a strong natural advantage for sustaining enset-based farming systems, which are crucial for food security and local livelihoods (see Fig 5).

**Slope.** Slope, the measure of land steepness, plays a vital role in agricultural suitability. Gentle or flat slopes are easier to cultivate and retain nutrients, while steeper slopes pose challenges due to erosion and nutrient loss. In this study, slope data were extracted from a 30 m resolution SRTM Digital Elevation Model (DEM) and reclassified into suitability categories to reflect different levels of agricultural potential, in line with [21].

Fig 6 presents the land suitability map derived from slope data (in degrees), illustrating the influence of topography on enset cultivation within the study area. The map reveals that a substantial portion of the Hadiya Zone is favorable for enset production. This classification is considered reliable as it incorporates key physical and environmental factors in the evaluation process. Suitability was determined by integrating climatic, soil, and topographic parameters to provide a comprehensive assessment in line with [16]. The spatial distribution of enset suitability classes is clearly depicted, showing that 21.2% of the area falls within the most suitable category (S1), 34.8% is moderately suitable (S2), 12.5% is marginally suitable (S3), and the remaining 10.8% and 20.7% is classified as not suitable (N1 and N2), see Table 6.

## Enset suitability analysis based on soil properties

**Soil types.** Soil type is one of the most important factors in determining land suitability for crops, and it was a major criterion in mapping enset suitability in the Hadiya Zone. Using soil data from the Ministry of Agriculture, classified under the FAO Soil Database; Field soil survey [12], and processed at 10 m resolution in ArcGIS, soils were evaluated based on FAO guidelines and expert judgment. The results show in Table 7 and Fig 7 blow that Nitisols and Andosols (16.8%) are highly suitable for enset cultivation, Vertisols, Luvisols and Fluvisols (55.8%) are moderately suitable, while Orthic Solonchaks (11.5%) are unsuitable.

**Soil texture.** Soil texture refers to the relative proportion of sand, silt, and clay in the soil, which determines its coarseness and water-holding capacity. Enset, being a tropical crop, thrives in soils that can retain moisture for extended periods. In this study, soil texture was considered an important criterion for assessing land suitability for enset cultivation. The soil texture dataset, obtained for the study area, was resampled to a 250m spatial resolution using the ArcGIS 10.8 Resample tool and then reclassified into five suitability classes ranging from highly suitable to unsuitable based on the crop's texture requirements.

**Table 5. Elevation suitability class of the study area.**

| Elevation classes | Level of suitability | Value | Area (Km²) | Area coverage (%) |
|---|---|---|---|---|
| >2,100m | Highly Suitable | 1 | 1079.3 | 29.7 |
| 2,100–1,900m | Moderately Suitable | 2 | 1621.5 | 44.6 |
| 1,900- 1700 | Marginally Suitable | 3 | 523.8 | 14.4 |
| 1700−1500 | Currently not Suitable | 1 | 192.4 | 5.3 |
| <1500 | Permanently not Suitable | 2 | 217.5 | 6.0 |

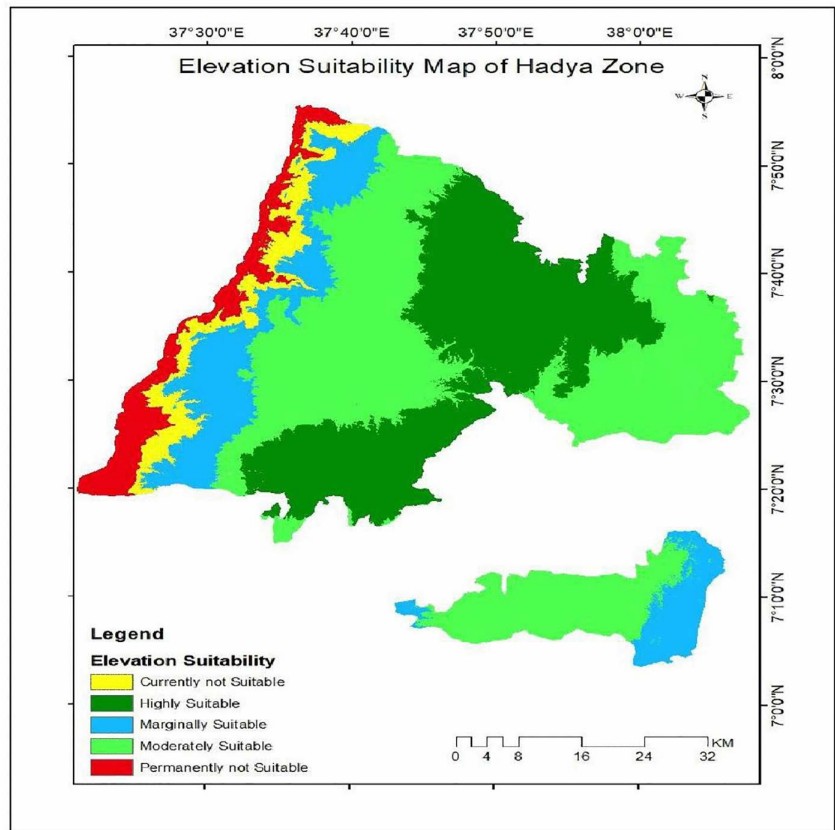

**Fig 5. Elevation suitability class of the study area.**

The results Table 8 and Fig 8 indicate that loam and clay loam soils, classified as highly to moderately suitable for enset cultivation, cover 3,010.5 km² (82.8%) of the total study area. Sandy clay and sandy clay loam soils, which are considered marginally suitable, occupy 156.3 km² (4.3%). Clay soils, currently classified as not suitable, account for 458.1 km² (12.6%). Only 10.1 km² (0.3%) of the area predominantly characterized by sandy clay soils is deemed unsuitable for enset cultivation.

**Soil depth.** The analysis results Table 9 and Fig 9 show that the vast majority of the study area, 3,616.5 km² (97.2%), is characterized by soil depths greater than 1 m, which are classified as highly suitable for enset cultivation. Such deep soils provide favorable conditions for root development, moisture retention, and nutrient availability, thereby supporting optimal enset growth and productivity (Mekonnen et al., 2020). Areas with soil depths ranging from 0.75 to 1 m account for 2.3% of the total area and are considered moderately suitable, while soils with depths of 0.5–0.75 m represent only 0.2% and are marginally suitable. Soil depths of less than 0.5 m, which limit root penetration and water storage, are classified as unsuitable for enset cultivation and cover only a negligible proportion of the study area.

**Soil organic carbon.** Soil Organic Carbon (SOC) plays a vital role in determining soil fertility, structure, and moisture-holding capacity, all of which are essential for optimal enset growth. Higher SOC levels enhance nutrient availability and resilience against environmental stress, making it a key criterion in land suitability evaluation for enset cultivation. The analyses result are in Table 10 and Fig 10 indicating that in the study area, 22.52% is highly suitable, 75.07% is moderately suitable, 1.58% is marginally suitable and 0.83% which is said to be currently not suitable.

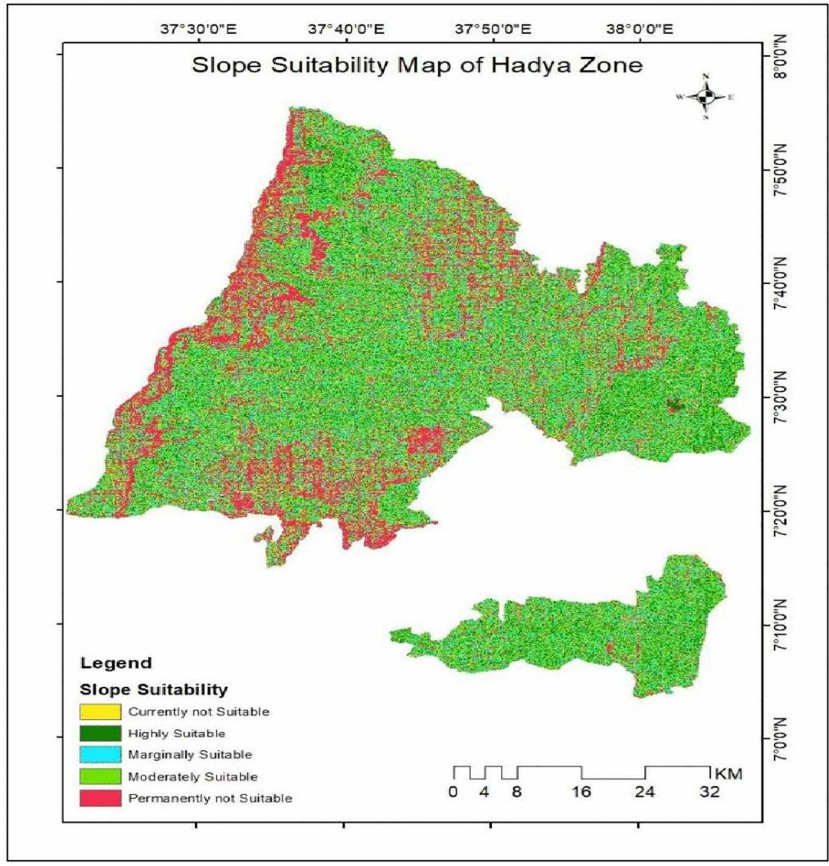

**Fig 6. Slope suitability class of the study area.**

**Table 6. Slope suitability class of the study area.**

| Range | Level of suitability | Value | Area Km² | Area coverage (%) |
|---|---|---|---|---|
| 1-4% | Highly Suitable | S1 | 770.75 | 21.2 |
| 4–8% | Moderately Suitable | S2 | 1265.2 | 34.8 |
| 8–12% | Marginally Suitable | S3 | 454.5 | 12.5 |
| 12 −15% | Currently not Suitable | N1 | 392.6 | 10.8 |
| >15% | Permanently not Suitable | N2 | 752.5 | 20.7 |

**Soil pH.** In the study area, soil pH values range from a minimum of 5.2 to a maximum of 7.1. The soil pH suitability map Fig 11 was derived by reclassifying the pH values into five suitability categories based on the FAO land evaluation framework and the optimal pH requirements for enset cultivation.

From the total area of the district (175.7 km²), 4.8% was highly suitable for enset cultivation regarding soil pH. Of the total study area (1006 km²), 27.7 km² was moderately suitable for enset crop cultivation, (915.7 km²) 25.2% was marginally suitable, (901.67 km²) 24.5% was currently not suitable, whereas 6467 km²) 17.8% of the study area was permanently not suitable for enset crop cultivation (see Table 11 and Fig 11).

**Table 7. Soil type suitability class of the study area.**

| Range | Level of suitability | Value | Area Km² | Area coverage (%) |
|---|---|---|---|---|
| Nitisols & Andosols | S1 | 1 | 609.1 | 16.8 |
| Vertisols, Luvisols & Fluvisols | S2 | 2 | 2027.3 | 55.8 |
| Cambisols & Nitisols (Dystric) | S3 | 3 | 498.3 | 13.7 |
| Leptosols & Calcisols/Xerosols | NI | 1 | 80.7 | 2.2 |
| Orthic Solonchaks | N2 | 2 | 419.8 | 11.5 |

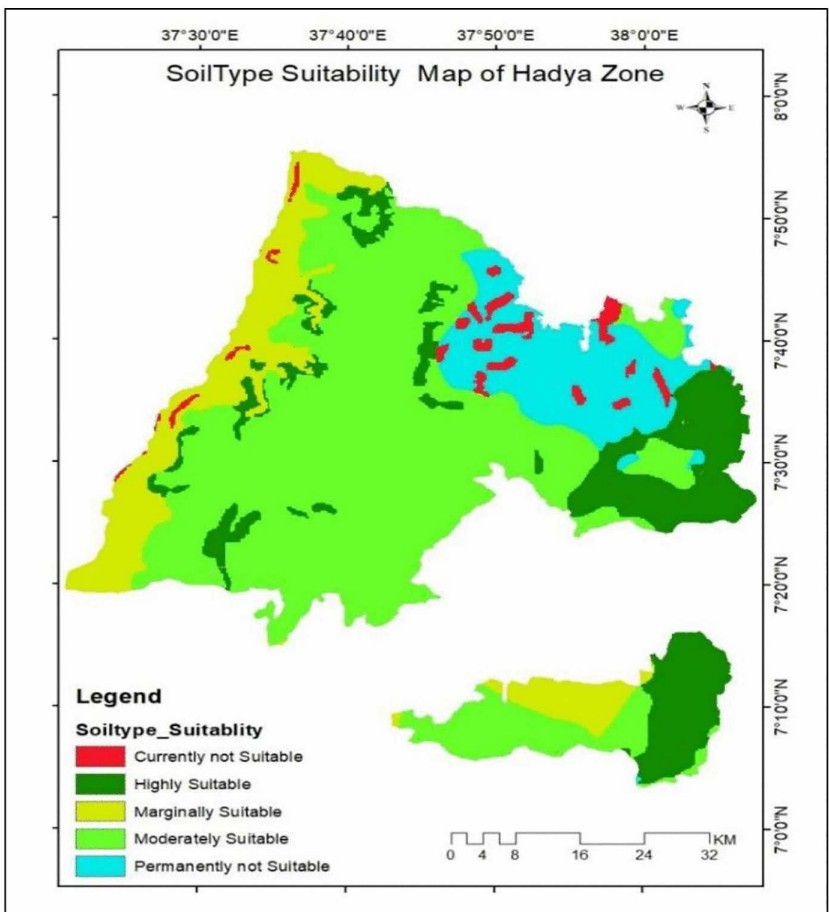

**Fig 7. Soil type suitability class of the study area.**

## Enset suitability analysis based on climate

Since the agricultural activity in the study area primarily relies on rain-fed systems, rainfall distribution plays a crucial role in determining agricultural suitability.

## Temperature suitability for enset cultivation

Based on the temperature requirements of enset cultivation, the thermal regime of the study area was reclassified into suitability classes using a land suitability analysis framework. As presented in Table 12 and Fig 12 three

**Table 8. Soil texture suitability class of the study area.**

| Range | Level of suitability | Value | Area Km² | Area coverage (%) |
|---|---|---|---|---|
| Loam | S1 | 1 | 383.4 | 10.5 |
| Clay Loam | S2 | 2 | 2627.2 | 72.3 |
| Sandy Loam & Sandy Clay Loam | S3 | 3 | 156.3 | 4.3 |
| Clay | N1 | 1 | 458.1 | 12.6 |
| Sandy Clay | N2 | 2 | 10.1 | 0.3 |

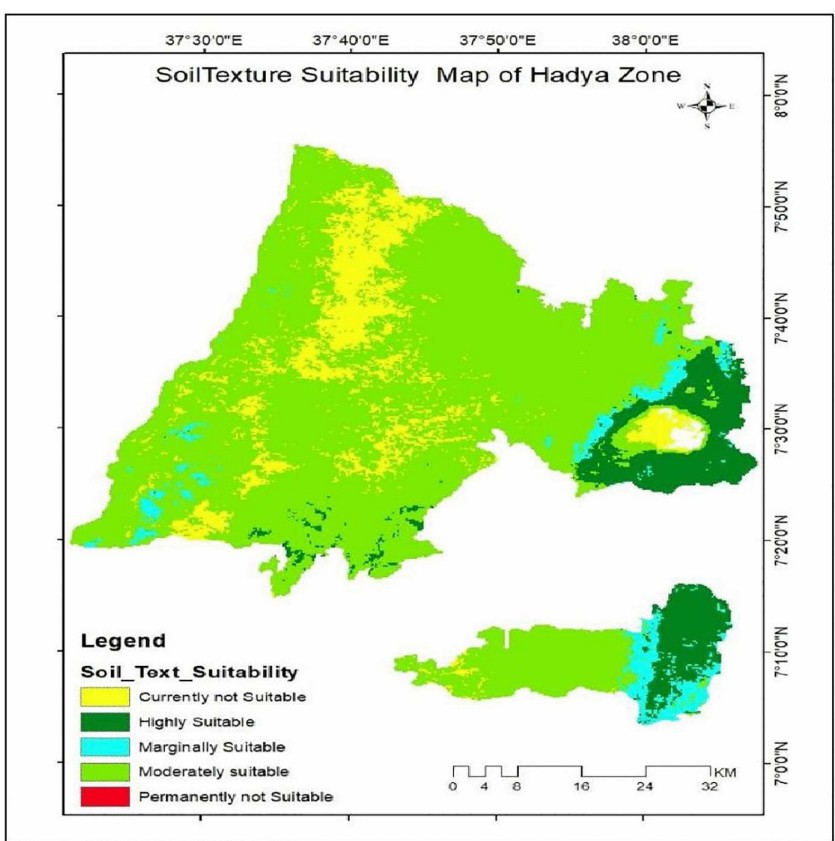

**Fig 8. Soil texture suitability class of the study area.**

**Table 9. Soil depth suitability class of the study area.**

| Range | Level of suitability | Value | Area Km² | Area coverage (%) |
|---|---|---|---|---|
| >1m | S1 | 1 | 3532.7 | 97.2 |
| 0.75–1m | S2 | 2 | 82.8 | 2.3 |
| 0.5–0.75m | S3 | 3 | 8.2 | 0.2 |
| 0.45–0.5m | N1 | 1 | 10.1 | 0.3 |
| <0.45m | N2 | 2 | 2.1 | 0.1 |

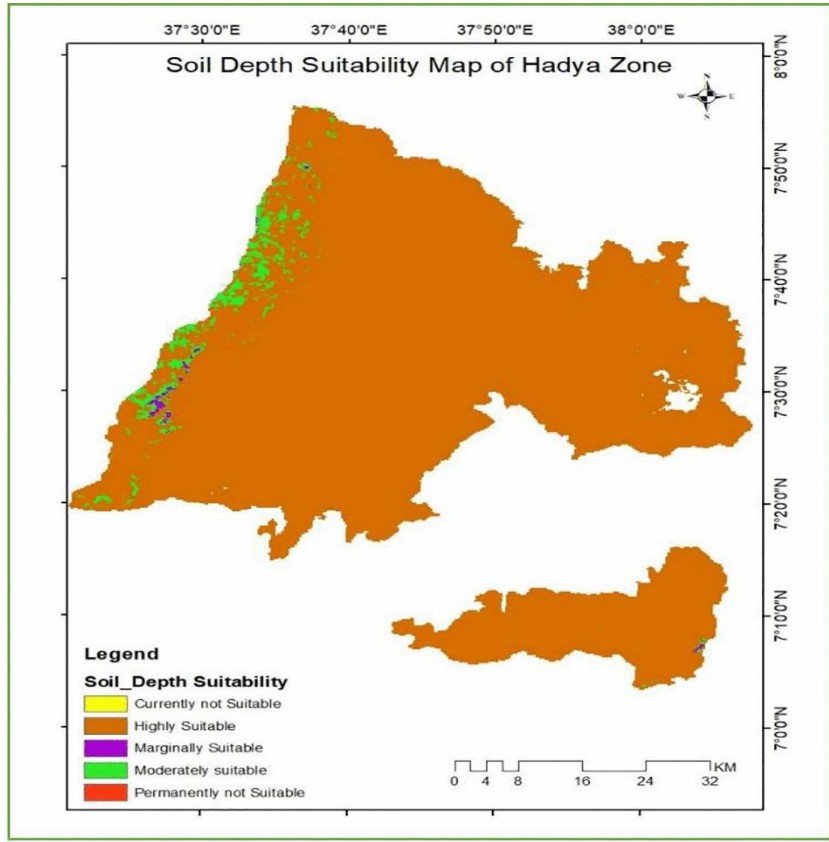

**Fig 9. Soil depth suitability class of the study area.**

**Table 10. Enset Suitability based on soil organic carbon.**

| Range | Level of suitability | Value | Area Km² | Area coverage (%) |
|-------|---------------------|-------|----------|-------------------|
| > 2.5 | S1 | 1 | 818.90 | 22.52 |
| 1.5–2.5 | S2 | 2 | 2729.2 | 75.07 |
| 0.8–1.5 | S3 | 3 | 57.4 | 1.58 |
| < 0.8 | N1 | 1 | 30.0 | 0.83 |

temperature-based suitability categories were identified: highly suitable (S1), moderately suitable (S2), and marginally suitable (S3). Temperature ranges of 15.5–16.5 °C, 16.5–18.5 °C, and 18.5–20.5 °C were classified as highly, moderately, and marginally suitable for enset cultivation, respectively. These classes account for 7.8% (S1), 83.8% (S2), and 8.4% (S3) of the total study area, indicating that the region is predominantly characterized by favorable thermal conditions for enset production, with the majority of the landscape falling within the optimal and moderately suitable temperature ranges.

**Source:** Climate data obtained from the World Bank Climate Knowledge Portal, derived from publicly available global and national climate datasets, provided under open-access data use policies. Temperature suitability classes (S1–S3) were generated using a land suitability analysis framework. Map generated and adapted by the authors.

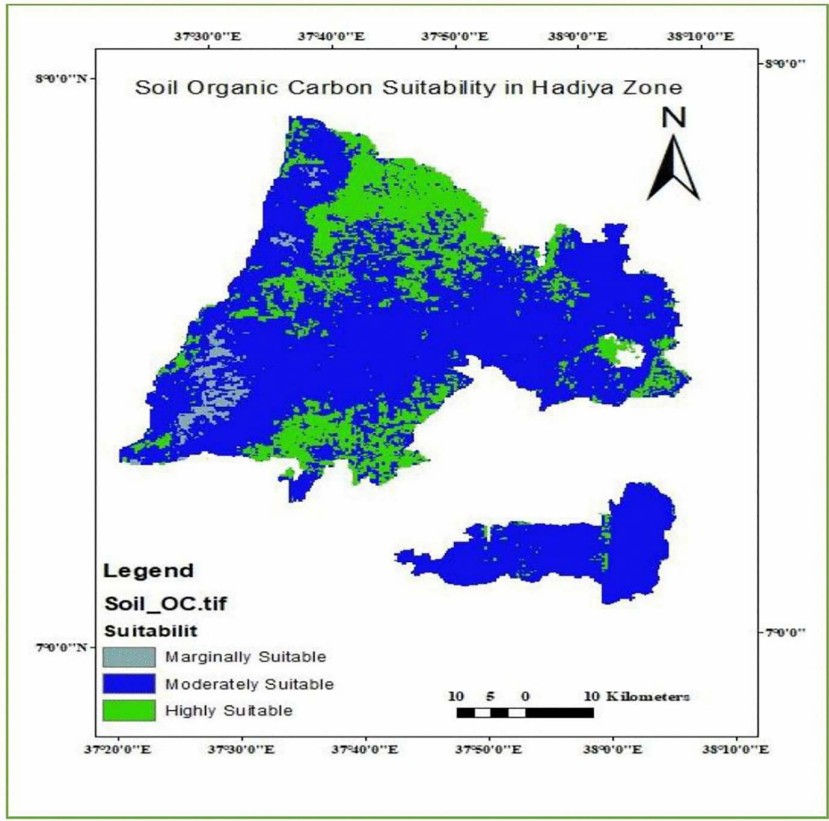

**Fig 10. Soil Organic Carbon Suitability Map of the study area.**

### Rainfall suitability for enset cultivation

As indicated in Table 13 and Fig 13 the rainfall distribution in the Hadiya Zone exhibits considerable spatial variability across the study area. Areas receiving more than 1350 mm of mean annual rainfall cover approximately 160 km² (4.4%) of the total land area. The largest proportion of the zone falls within the 1150–1350 mm rainfall range, encompassing about 2,857.85 km² (78.6%), indicating that the majority of the region experiences moderately high and reliable rainfall conditions. Areas receiving 950–1150 mm of rainfall account for 551.33 km² (15.2%), while the lowest rainfall class (800–950 mm) covers only 66.44 km² (1.8%) of the total area. This spatial distribution pattern demonstrates that most of the Hadiya Zone is characterized by favorable rainfall regimes suitable for mixed farming systems, including enset cultivation, which performs optimally under conditions of moderately high, well-distributed, and reliable rainfall.

**Source:** Climate data obtained from the World Bank Climate Knowledge Portal, derived from publicly available global and national climate datasets, provided under open-access data use policies. Rainfall suitability classes were generated using a land suitability analysis framework. Map generated and adapted by the authors.

### Calculating the overall suitability for enset cultivation

We developed the suitability classification for each parameter that was used in evaluating the potential of the study region for enset cultivation by conducting a systematic comparison between the spatial distribution of each environmental element and the established enset crop requirement standards. This comparison was utilised to determine whether or not the study area was suitable for enset cultivation. Temperature, rainfall, soil depth, soil texture, soil type, soil organic carbon

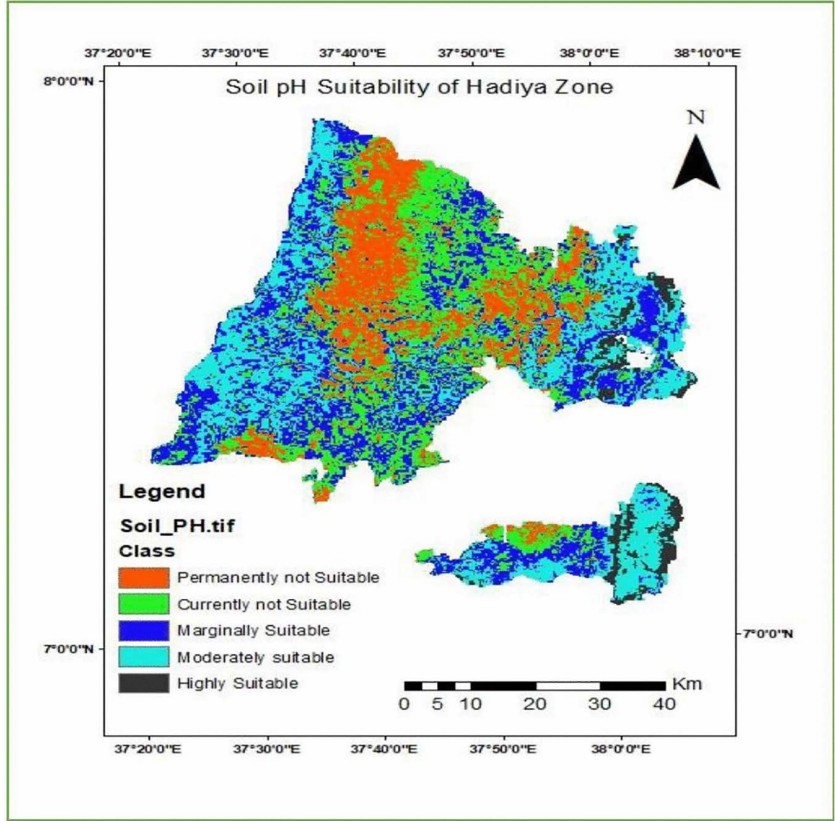

**Fig 11. Enset crop cultivation Suitability map based on soil pH.**

**Table 11. Soil pH suitability class of the study area.**

| Soil pH | Level of suitability | Value | Area (Km2) | Area (%) |
|---|---|---|---|---|
| 6.5-7.1 | Highly Suitable | 1 | 175.7 | 4.8 |
| 6.0-6.5 | Moderately Suitable | 2 | 1006 | 27.7 |
| 5.6-6.00 | Marginally Suitable | 3 | 915.7 | 25.2 |
| 5.4-5.6 | Currently Not Suitable | 1 | 901.6 | 24.5 |
| <5.4 | Permanently not suitable | 2 | 646.0 | 17.8 |

**Table 12. Temperature suitability class.**

| Temperature (oC) | Level of suitability | Value | Area (km²) | Area coverage (%) |
|---|---|---|---|---|
| 15.5-16.5°C | S1 | 1 | 282.6 | 7.8 |
| 16.5-18.5.°C | S2 | 1 | 3046.3 | 83.8 |
| 18.5-21.5°C | S3 | 3 | 306.7 | 8.4 |

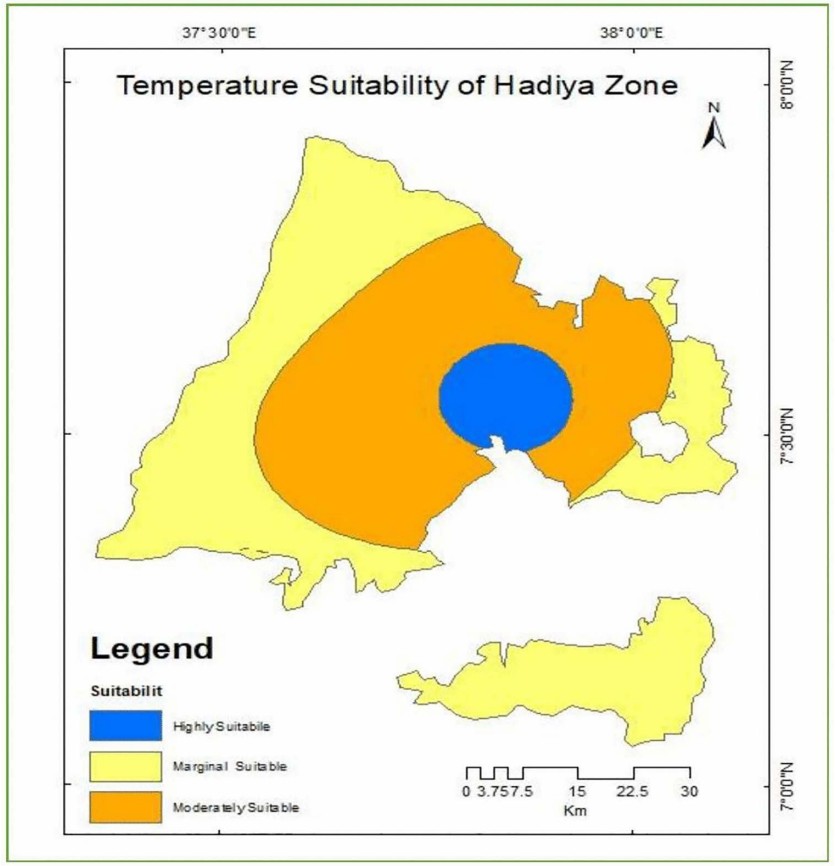

**Fig 12. Land suitability classes for enset cultivation based on Temperature.**

**Table 13. Land suitability classes for enset cultivation based on Rainfall.**

| Rainfall (mm) | Level of suitability | Value | Area (km²) | Area coverage (%) |
|---|---|---|---|---|
| >1350 | S1 | 1 | 160 | 4.4 |
| 1150-1350 | S2 | 2 | 2857.85 | 78.6 |
| 950-1150 | S3 | 3 | 551.32 | 15.2 |
| 800-950 | N1 | 1 | 66.44 | 1.8 |

(SOC), soil pH, elevation, slope, and land use/land cover (LULC) were some of the elements that were taken into consideration during this investigation.

For every parameter, the relevant values within the research region were overlaid with ecological thresholds that were specific to the enset. These criteria were obtained from academic literature and conversations with specialists. According to [11] land evaluation framework, each factor was classified into one of five suitability classes based on these comparisons. These grades were as follows: Very Highly Suitable (S1), Highly Suitable (S2), Moderately Suitable (S3), Marginally Suitable (N1), and Permanently Not Suitable (N2). The detection of geographical variations in suitability for enset cultivation was made possible by this classification, which also served as a platform for further multi-criteria evaluation and weighted overlay analysis.

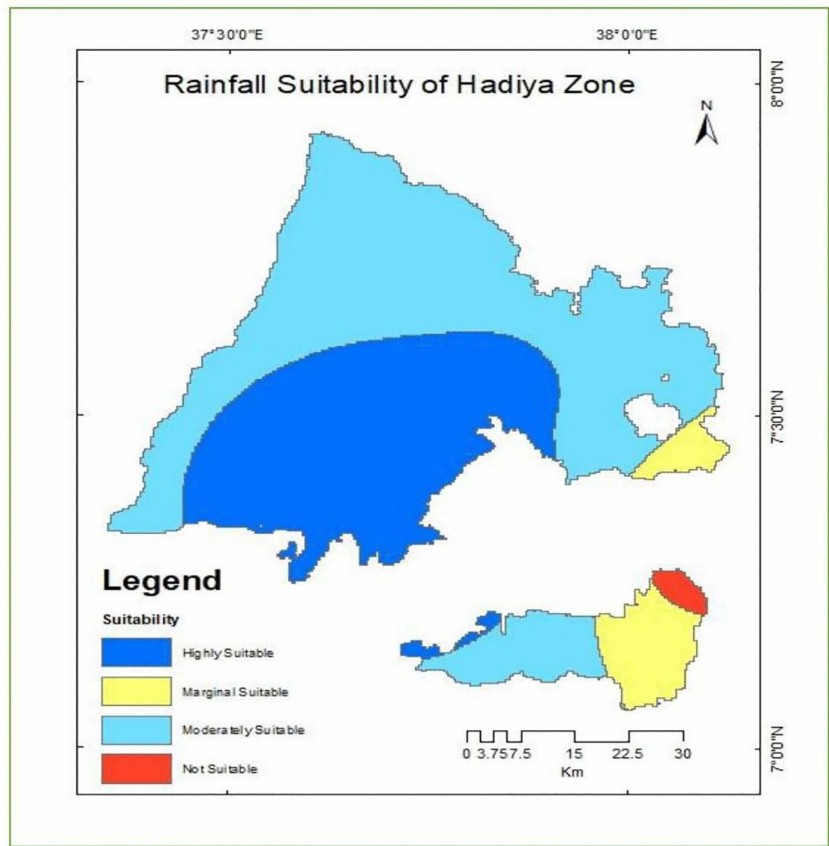

**Fig 13. Land suitability classes for enset cultivation based on Rainfall.**

As indicated in Table 14 and Fig. 14 the integrated analysis of soil, climate, topography, and land use shows that most of the Hadiya Zone is favorable for enset cultivation. About 57.72% (3,635.5 km$^2$) is highly suitable (S1), 36.89% is moderately suitable (S2), 0.159% is marginally suitable (S3), and 5.231% is currently not suitable (N1), with no areas permanently unsuitable (N2). Overall, the findings highlight the zone's strong natural potential for enset, though localized challenges such as soil fertility, water management, or slope issues may require targeted interventions. This dominance of highly suitable land reinforces enset's importance as a staple crop for the region.

## Summary, conclusion and recommendation

**Summary and conclusion.** This research was intended to evaluate the physical land suitability for the enset crop by integrating GIS and Remote Sensing with Multi-criteria Evaluation in the Hadiya zone. Land suitability evaluation for agriculture is a very important piece of information for agricultural development and future planning. Land suitability analysis is a complex process and includes various domains of knowledge. The parameters used for this suitability analysis were topography (slope, elevation), soil (type, texture, pH, OC and depth), Climate (temperature and rainfall) and LULC. The land evaluation of the physical land qualities of the study area indicates that the zone has great potential for enset cultivation. The FAO land evaluation method is used in evaluating the suitability of the area for enset cultivation. Based on the findings from the parameters used in the study, elevation, rainfall, and temperature are dominant factors that influence the suitability of land for enset crop cultivation in the study area, rather than the other factors.

**Table 14. The area of each factor with its level of suitability.**

| Factor | Suitability Class | | | | | Total Area km² | Influencing rate |
|---|---|---|---|---|---|---|---|
| | Highly Suitable% | Moderately Suitable% | Marginally Suitable% | Currently not Suitable % | Permanently not Suitable % | | |
| Elevation | 29.7 | 44.6 | 15.4 | 5.3 | 6.0 | 3635.5 | 28 |
| LULC | 52.0 | 15.4 | 10.6 | 20.0 | 1.7 | 3635.5 | 9 |
| Slope | 16.2 | 29.8 | 22.5 | 10.8 | 20.4 | 3635.5 | 3 |
| Soil depth | 96.8 | 2.3 | 0.2 | 0.3 | 0.1 | 3635.5 | 6 |
| Soil Ph | 4.8 | 24.9 | 26.6 | 26.5 | 17.8 | 3635.5 | 3 |
| Soil OC | 22.5 | 75.4 | 2.5 | ---- | ---- | 3635.5 | 3 |
| Soil texture | 10.5 | 72.3 | 4.7 | 12.6 | 0.3 | 3635.5 | 6 |
| Soil Type | 16.8 | 55.8 | 13.7 | 2.2 | 11.8 | 3635.5 | 8 |
| Temperatur | 7.8 | 83.8 | 8.3 | ---- | ---- | 3635.5 | 15 |
| Rainfall | 4.4 | 78.6 | 15.2 | 1.9 | ---- | 3635.5 | 19 |
| **Overall suitability** | **57.72** | **36.89** | **0.159** | **5.231** | **–** | **3,635.5** | |

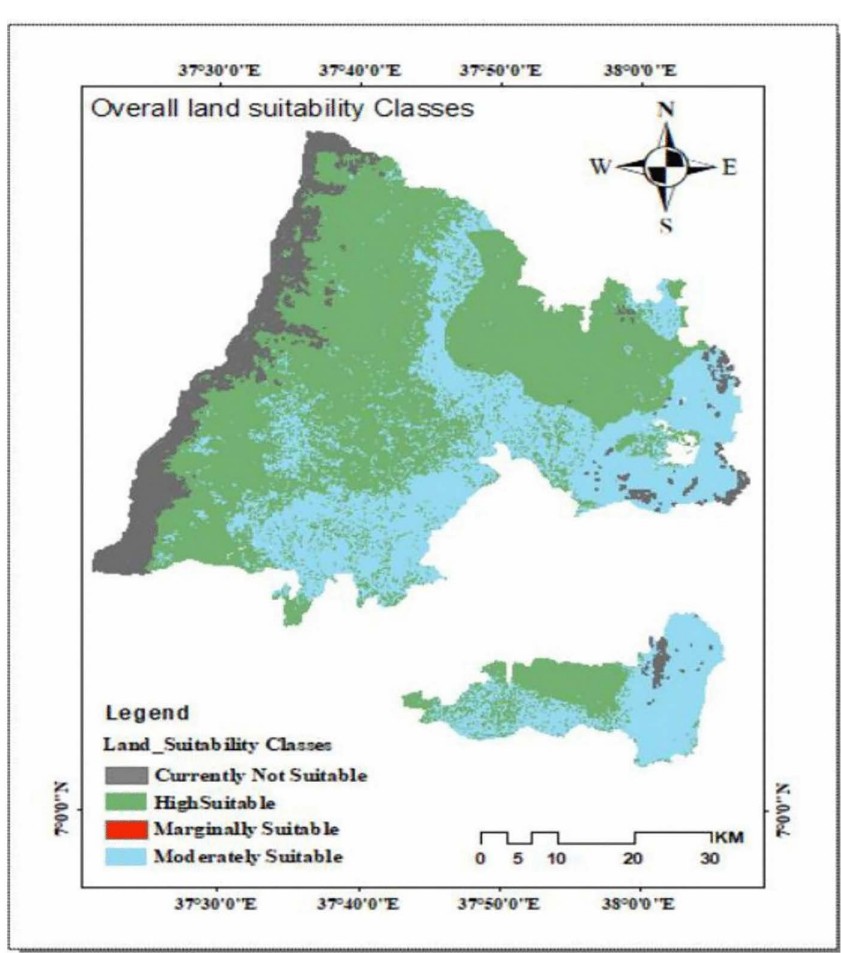

**Fig 14. Overall suitability map of the Hadiya zone for enset cultivation.**

The land suitability of the study area is classified into five (S1, S2, S3, N1 &N2). In terms of soil depth, most of the crops require deep soil depth. Enset is one of the crops that requires deep soil. In the study area, 97.2% of % soil depth is > 1meter, which means the study area is very suitable for enset cultivation. About 46% of the zone is characterized by a gentle slope, which is less than 8%. Generally, the results showed that 2088.79 km$^2$ (57.72%), 1339.81 km$^2$ (36.89%), 19.0 km$^2$ (0,159%) and 188.21 km$^2$ (5.231%) of the total area were highly suitable, moderately suitable, marginally suitable and currently not suitable, respectively, for enset cultivation in the study area.

**Recommendations.**

- The land use types examined in this study were restricted to a single perennial crop. To broaden the alternatives available for land-use planning, further research should include other land use types, such as additional perennials and cereals, to identify the most suitable and sustainable options for specific parcels of land.

- The study area in Central Ethiopia relies heavily on enset farming, yet lacks a dedicated research center to address its major challenges. Critical issues such as disease management, yield improvement, clone selection, optimal spacing, transplanting, fertilizer use, and the long maturation period remain poorly studied. Establishing such a research centre is essential to enhance enset productivity and sustainability.

- This study assessed land suitability for enset cultivation based solely on physical parameters, including soil properties (depth, texture, pH, SOC, and type), climate factors (temperature and rainfall), land use/land cover, elevation, and slope. While these provide valuable insights, the analysis could be further enriched by integrating socio-economic variables such as farmers' practices, access to markets, and labour availability, which play a crucial role in shaping the actual suitability and long-term sustainability of enset cultivation.

- Although the zone is highly suitable for enset cultivation, certain challenges, especially low soil pH and sloping land, need careful management. To enhance productivity and profitability, farmers and stakeholders should focus on correcting soil pH and applying soil and water conservation practices custom-made to slope conditions.

- As only a small portion of the land is classified as unsuitable, the majority of the area is highly suitable for enset cultivation. Therefore, concerned stakeholders are encouraged to promote and invest in enset production, recognising its comparative advantages over other crops, particularly its significant role in environmental conservation and its nutritional and livelihood value for both humans and livestock.

- Enset is a major staple crop in the study area, with productivity levels that exceed most other crops. Owing to its high yield potential and multiple benefits, smallholder farmers are strongly encouraged to strengthen and expand enset cultivation practices. Promoting enset farming not only improves household food security and income but also supports sustainable land management and enhances long-term resilience in the area.

## Acknowledgments

The authors gratefully acknowledge the Department of Geography, Andhra University, for their invaluable support in facilitating this research on enset in Ethiopia. We also extend our sincere appreciation to Wachemo University for their cooperation and continuous encouragement throughout the study. Finally, our heartfelt thanks go to all respondents and key informants in the Hadiya Zone for generously sharing their time and insights.

## Author contributions

**Conceptualization:** Alemu Ersino Ersado, Venkata Krishna Talluri.

**Data curation:** Alemu Ersino Ersado.

**Formal analysis:** Alemu Ersino Ersado.

**Funding acquisition:** Alemu Ersino Ersado.

**Investigation:** Alemu Ersino Ersado.

**Methodology:** Alemu Ersino Ersado, Venkata Krishna Talluri.

**Project administration:** Venkata Krishna Talluri.

**Resources:** Alemu Ersino Ersado, Venkata Krishna Talluri.

**Software:** Alemu Ersino Ersado.

**Supervision:** Venkata Krishna Talluri.

**Validation:** Venkata Krishna Talluri.

**Visualization:** Alemu Ersino Ersado, Venkata Krishna Talluri.

**Writing – original draft:** Alemu Ersino Ersado.

**Writing – review & editing:** Alemu Ersino Ersado.

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
