## [Decision Letter · Decision Letter 0]

4 Nov 2025

Dear Dr. Ersado,

Thank you for submitting your manuscript to PLOS ONE. After careful consideration, we feel that it has merit but does not fully meet PLOS ONE’s publication criteria as it currently stands. Therefore, we invite you to submit a revised version of the manuscript that addresses the points raised during the review process.

We look forward to receiving your revised manuscript.

Kind regards,

Pradeep Kumar Badapalli

Academic Editor

PLOS ONE

Journal Requirements:

4. In the online submission form, you indicated that “All data generated or analyzed during this study are included in this published article. Additional data can be provided by the corresponding author upon reasonable request.”

5. We note that your Data Availability Statement is currently as follows: “All data generated or analyzed during this study are included in this published article. Additional data can be provided by the corresponding author upon reasonable request.”

6. Please ensure that you refer to Figure 1,2,4,5,6,9,11,12,13 and 14 in your text as, if accepted, production will need this reference to link the reader to the figure.

7. Please include a caption for figure 14.

8. We note that Figures 1 and 4 - 14 in your submission contain satellite images which may be copyrighted. All PLOS content is published under the Creative Commons Attribution License (CC BY 4.0), which means that the manuscript, images, and Supporting Information files will be freely available online, and any third party is permitted to access, download, copy, distribute, and use these materials in any way, even commercially, with proper attribution.

For these reasons, we cannot publish previously copyrighted maps or satellite images created using proprietary data, such as Google software (Google Maps, Street View, and Earth). For more information, see our copyright guidelines: http://journals.plos.org/plosone/s/licenses-and-copyright.

1. You may seek permission from the original copyright holder of Figure(s) [#] to publish the content specifically under the CC BY 4.0 license.

9. We note you have included a table to which you do not refer in the text of your manuscript. Please ensure that you refer to Table 1,3,4,5,6,10,11,12 and 14 in your text; if accepted, production will need this reference to link the reader to the Table.

Additional Editor Comments:

Dear Authors,

Thank you for your interest in submitting your work to our journal. I have now received the reviewers’ reports, and based on their feedback, I have decided to request Major Revisions.

Please ensure that you address each and every comment raised by the reviewers and prepare a detailed rebuttal document explaining your responses.

Failure to adequately address the reviewers’ queries may affect the final decision on your manuscript.

Thank you for your understanding and cooperation.

Reviewers' comments:

Reviewer's Responses to Questions

**Comments to the Author**

1. Is the manuscript technically sound, and do the data support the conclusions?

Reviewer #1: Yes

Reviewer #2: Yes

2. Has the statistical analysis been performed appropriately and rigorously?

Reviewer #1: Yes

Reviewer #2: Yes

3. Have the authors made all data underlying the findings in their manuscript fully available?

Reviewer #1: Yes

Reviewer #2: Yes

4. Is the manuscript presented in an intelligible fashion and written in standard English?

Reviewer #1: Yes

Reviewer #2: No

Reviewer #1: 1. What was the selection of criteria or factors based on? Please explain.

2. In the discussion section, it is better to make comparisons with the researches of others and the research done, and for the presented arguments, be sure to use valid and up-to-date references.

3. Please provide valid references for the relationships provided.

4. Why have properties such as lime, gypsum, and exchangeable sodium not been examined in examining soil properties?

5. The advantages and disadvantages of the research done should be said.

6. In the AHP method, did you use the opinions of relevant experts in the form of a questionnaire to determine the importance of the factors in the pairwise comparison matrix?

7. Please use the papers (https://doi.org/10.1007/s10661-022-10327-x,
https://doi.org/10.1080/03650340.2018.1549363,
https://doi.org/10.1016/j.geoderma.2017.09.012, https://doi.org/10.1016/j.geoderma.2019.05.046, https://doi.org/10.1080/00103624.2022.2072511; https://doi.org/10.1080/03067319.2020.1746775; https://doi.org/10.1007/s10661-022-10659-8; https://doi.org/10.1080/00103624.2019.1626870) to improve the quality of the manuscript and add them to the manuscript, especially the introduction and discussion of the manuscript, description and interpretation of properties and select the criteria.

8. Please give the names of soils according to the WRB system.

9. What was the accuracy of the methods used? By which criteria are the methods evaluated?

10. Please check the grammar of the whole text with a native speaker and fix the errors.

Reviewer #2: General Overview

This manuscript presents a GIS-based Multi-Criteria Decision Analysis (MCDA) for evaluating land suitability for enset cultivation in the Hadiya Zone, Ethiopia. The study is highly relevant, addresses a clear research gap, and employs a robust methodology following the FAO framework. The findings are significant, indicating that a large majority of the zone is highly or moderately suitable for enset, with climate and topography being the dominant influencing factors.

Major Strengths

The introduction effectively establishes the importance of enset for food security in Ethiopia and the need for scientific land suitability analysis to move beyond traditional knowledge.

The application of GIS, remote sensing, and the Analytical Hierarchy Process (AHP) is appropriate for the research question. The use of the FAO framework provides a solid theoretical foundation.

The study integrates a wide range of relevant data layers (soil, climate, topography, LULC), which is crucial for a holistic suitability assessment.

The calculation and reporting of the Consistency Ratio (CR=3.4%) for the AHP adds credibility to the weighting of criteria.

The conclusions are clear, and the recommendations are practical and targeted for farmers, planners, and policymakers.

Major Points for Revision and Clarification

1. Title and Abstract:

Title: There is a typo in the species name. It should be (Welw.) not (Webv.) or ((Ensete ventricosum (Webv.). Please correct to (Ensete ventricosum (Welw.) Cheesman).

Abstract & Manuscript: The term "onset" is used frequently instead of "enset" (e.g., Page 8, line 1; Page 10, Specific Objective 2). This is a critical error as it changes the subject of the paper. This must be corrected throughout the entire manuscript.

2. Methodology Section:

-Data Sources and Resolution: Table 1 is helpful but needs clarification. Some entries are confusing (e.g., Climate Data has "ERA5: 0.25° (~31 km), Raster (1 km²)" which one was used? DEM lists both ASTER 30m and SRTM 10m). Please specify the exact dataset and its final spatial resolution used for the analysis for each factor. Consistency in resolution is key for overlay analysis.

-Criterion Weighting: The process for obtaining the pairwise comparisons (the expert judgments) is not described. How many experts were involved? What was their field of expertise? Were they local agronomists, soil scientists, etc.? A brief description would strengthen this section.

-Suitability Classification Thresholds: The manuscript would be significantly strengthened by including a table or a section in the methodology that defines the thresholds used to classify each factor into S1, S2, S3, N1, N2. For example, what specific elevation, slope, pH, and SOC ranges define "Highly Suitable" for enset? Citing established literature (e.g., FAO, previous enset studies) for these thresholds is essential.

3. Results and Discussion Section:

Integration of Results:

The results are presented well for each individual factor, but the discussion of the integrated overall result (Section 3.3) is relatively brief. Expand this section.

Discuss any interesting spatial patterns in the final map (Fig. 13). Where are the S1 areas concentrated? Do they correlate with specific districts?

Discuss potential trade-offs. For example, an area might have highly suitable soil but a less suitable slope. How does the model handle this?

Validation:

The study lacks a validation of the final suitability map. How can we be confident that the areas classified as "Highly Suitable" actually yield better enset? Consider suggesting this for future work, or if possible, perform a simple validation by comparing your map with known high-yield enset areas or through ground-truthing surveys.

Inconsistency in Final Area Calculation:

On Page 24, Table 14 and the text state the total area is 3,635.5 km², with 57.7% (2,098.79 km²) as S1.

On Page 25, the text states "2088.79 km² (57.2%)". There is a discrepancy between the percentage (57.7% vs. 57.2%) and the calculated area. Please recalculate and ensure all area figures and percentages are consistent throughout the manuscript.

4. Figures:

The figures referenced in the text (Fig. 1, 2, 3, etc.) are not included in the provided PDF in a viewable format. The authors must ensure all figures are of high quality, with legible legends, scales, and north arrows.

Figure 13 (Overall Suitability Map): The caption in the image says, "THE OVERALL SAUTABILITY" please correct to "SUITABILITY". The map legend is also cut off. Ensure the final version is clear and complete.

Minor Corrections and Suggestions

Page 9, Introduction: "analysis of land suitability is very needed" "is highly necessary" or "is essential".

Page 10, Specific Objectives: The second objective ("examine the major factors...") is partially methodological and seems to be answered by the AHP weights. Consider rephrasing to better align with the mapping focus, e.g., "To quantify the relative influence of different factors on enset cultivation suitability."

Page 12, Section 2.1.1. Astronomic location: The term is typically "Geographic location" or "Astronomical location". "Astronomic" is uncommon. The latitude is given as 7°07' - 7°92' N. This is likely a typo, as 92' is not possible (60 minutes = 1 degree). It should probably be 7°07' - 7°52' N or similar. Please verify the coordinates.

Page 23, Temperature Section: There is a copy-paste error: "was classified as highly and moderately suitable, respectively, for rice crop cultivation". This should be corrected to "enset".

Page 25, Summary: "97.2% of % soil depth" "97.2% of the area has a soil depth".

References: The reference list is extensive, but some formatting is inconsistent (e.g., use of italics, capitalization). Please ensure it strictly adheres to the PLOS ONE reference style guide.

**Do you want your identity to be public for this peer review?** For information about this choice, including consent withdrawal, please see our Privacy Policy

Reviewer #1: **Yes:** Javad seyedmohammadi

Reviewer #2: No

---

## [Author Response · Author response to Decision Letter 1]

25 Nov 2025

I am grateful for your message and for the opportunity to provide some light on the matter. I agree with the fact that the publishing article contains all of the data that was gathered and examined throughout the process of this research. The authors of this study gathered all of the measurements, satellite images, and field observations that were used to compile the datasets that were utilized in this research. All of the relevant information has been thoroughly evaluated, interpreted, and presented in the sections of the text that are devoted to the results and the discussion.

---

## [Decision Letter · Decision Letter 1]

21 Dec 2025

Dear Dr. Ersado,

Thank you for submitting your manuscript to PLOS ONE. After careful consideration, we feel that it has merit but does not fully meet PLOS ONE’s publication criteria as it currently stands. Therefore, we invite you to submit a revised version of the manuscript that addresses the points raised during the review process.

We look forward to receiving your revised manuscript.

Kind regards,

Pradeep Kumar Badapalli

Academic Editor

PLOS One

**Journal Requirements:**

**Additional Editor Comments:**

The manuscript will be accepted after addressing the following minor corrections:

Adjust and align all figures properly.

Revise the methodology figure to improve clarity.

The figure is currently merged; please separate the components, use appropriate arrow symbols, and ensure a clean and logical workflow (refer and add: https://doi.org/10.1016/j.jenvman.2025.126820

; https://doi.org/10.1016/j.rsase.2025.101654).

Ensure uniform text style across all figures.

Remove figure headings and include them in the figure legends.

Ensure that all legends are clearly visible and readable.

Maintain consistency in equation numbering throughout the manuscript.

Reviewers' comments:

Reviewer's Responses to Questions

**Comments to the Author**

Reviewer #1: All comments have been addressed

Reviewer #2: All comments have been addressed

2. Is the manuscript technically sound, and do the data support the conclusions?

Reviewer #1: No

Reviewer #2: Yes

3. Has the statistical analysis been performed appropriately and rigorously?

Reviewer #1: N/A

Reviewer #2: Yes

4. Have the authors made all data underlying the findings in their manuscript fully available?

Reviewer #1: Yes

Reviewer #2: Yes

5. Is the manuscript presented in an intelligible fashion and written in standard English?

Reviewer #1: Yes

Reviewer #2: Yes

Reviewer #1: The respected authors of the article have corrected the comments point by point. The article is now sufficiently improved.

Reviewer #2: Thank you for the thoughtful revisions. I appreciate the work you've done to address my previous feedback it has significantly strengthened the manuscript.

**Do you want your identity to be public for this peer review?** For information about this choice, including consent withdrawal, please see our Privacy Policy

Reviewer #1: **Yes:** Javad seyedmohammadi

Reviewer #2: No

---

## [Author Response · Author response to Decision Letter 2]

25 Dec 2025

I would like to sincerely thank the editors and reviewers of PLOS ONE for their valuable time, constructive comments, and insightful suggestions. I have learnt greatly from your expertise throughout the review process. I truly appreciate every comment provided, and all points raised by the editors and reviewers have been carefully addressed. The detailed responses are provided below, point by point.

---

## [Editor Report · Decision Letter 2]

8 Jan 2026

Dear Dr. Ersado,

Thank you for submitting your manuscript to PLOS ONE. After careful consideration, we feel that it has merit but does not fully meet PLOS ONE’s publication criteria as it currently stands. Therefore, we invite you to submit a revised version of the manuscript that addresses the points raised during the review process.

We look forward to receiving your revised manuscript.

Kind regards,

Pradeep Kumar Badapalli

Academic Editor

PLOS One

**Journal Requirements:**

**Additional Editor Comments:**

Dear Authors,

The comments have been addressed; however, they remain somewhat limited.

Acceptance will be considered after improving the resolution of the images.

Please ensure that the figure legends are clear and that all text within the figures is clearly visible.

Kindly take care of these issues and resubmit the revised manuscript.

Thank you.

---

## [Author Response · Author response to Decision Letter 3]

29 Jan 2026

We are grateful for your insightful comments. We have carefully revised the manuscript and incorporated the suggested improvements. A point-by-point response addressing all comments raised by the academic editor and reviewers has been prepared and uploaded as a separate file titled “Response to Reviewers.

---

## [Editor Report · Decision Letter 3]

16 Feb 2026

GIS-Based Land Suitability Evaluation and Multi-Criteria Decision Analysis for Sustainable Enset (Ensete ventricosum (Welw.) Cheesman) Cultivation in Hadiya Zone, Central Ethiopia

PONE-D-25-54032R3

Dear Dr. Alemu Ersino Ersado

We’re pleased to inform you that your manuscript has been judged scientifically suitable for publication and will be formally accepted for publication once it meets all outstanding technical requirements.

Kind regards,

Pradeep Kumar Badapalli

Academic Editor

PLOS One

Additional Editor Comments (optional):

Dear Authors,

Based on the reviewer reports, the manuscript is now accepted for publication. Thank you for your interest in submitting your work to our journal.

---

## [Editor Report · Acceptance letter]

PONE-D-25-54032R3

PLOS One

Dear Dr. Ersado,

I'm pleased to inform you that your manuscript has been deemed suitable for publication in PLOS One. Congratulations! Your manuscript is now being handed over to our production team.

Kind regards,

on behalf of

Dr. Pradeep Kumar Badapalli

Academic Editor

PLOS One